# Erlang-U: Blocking Probability of UAV-Assisted Cellular Systems

**Mario E. Rivero-Angeles ***, **Iclia Villordo-Jimenez ***, **Izlian Y. Orea-Flores ***, **Noé Torres-Cruz *** and **Angel Pretelín Ricárdez ***

Computer Research Center, Instituto Politécnico Nacional (CIC-IPN), Ciudad de Mexico 07700, Mexico
* Correspondence: mriveroa@ipn.mx (M.E.R.-A.); ivillordo@ipn.mx (I.V.-J.); iorea@ipn.mx (I.Y.O.-F.); ntorresc@ipn.mx (N.T.-C.); apretelin@ipn.mx (A.P.R.)

**Abstract:** In modern and future communication systems, we expect peaks of traffic that largely exceed the capacity of the system, since they are originally designed to support normal traffic loads. Such peaks can be caused by emergency events and cultural or sporting gatherings, among others. Indeed, implementing more channels than the ones required in normal traffic conditions would entail higher costs and energy consumption. As such, when a traffic peak arrives, the system performance is greatly affected. To this end, we propose the use of mobile channels that assist cellular systems to increase the capacity of the network for a certain period. In this paper, we derive the blocking probability of a UAV (Unmanned Aerial Vehicle)-assisted cellular system to temporarily increase the capacity of the communication network in case of a traffic overload. The analysis presented in this work allows a careful design of future communication systems requiring fewer channels, that can serve users in normal traffic load conditions while using UAVs to maintain an adequate blocking probability when the traffic load increases. To this end, we develop the ErlangU formula, similar to the ErlangB formula for a *conventional* voice service cellular system.

**Keywords:** teletraffic analysis; blocking probability; UAVs; dropped calls

## 1. Introduction

In modern telecommunicatiions systems, the use of drones is becoming more and more relevant due to the flexibility and ease of deployment. For instance, in cellular systems, drones can assist with additional channels [1–11], and also in the Internet of Things (IoT) to charge nodes or recollect data sensed by nodes in a region of interest [12]. For these cases, it is preferable to send drones to the region of interest with additional channels, in the case of cellular systems, in a temporary manner rather than setting new base stations, especially if traffic peaks also occur in this manner, i.e., temporarily.

Cellular systems are usually designed to provide a blocking probability of 0.02 or 0.01 for voice services by carefully selecting the number of channels according to the peak traffic load in the coverage area of the cell. However, when certain events occur in this region, the traffic load can greatly increase, entailing a much higher blocking probability, which can affect the Quality of Service (QoS) of users. Events like emergencies (fires, earthquakes, flooding, etc.), cultural (concerts, festivals, etc.), sportive (games, Olympics, racing, etc.), and many others, are expected to temporarily increase the traffic load.

One alternative to tackling this problem is to provide additional channels in the Base Station. Still, the installation and operational costs are high and the extra channels would be active all the time, even when traffic returns to normal conditions. This would cause higher energy consumption in the overall system. Mobile base stations can also be implemented using vehicles. However, the advantage of using drones is that they are not constrained by terrestrial traffic nor by parking availability. Another important advantage of drones versus vehicles when introducing additional channels is that drones can be placed in the exact

region required to increase the capacity, for instance, in the case of a public event where streets are blocked to vehicle traffic. However, an important advantage of the use of vehicles is that the dwelling time is unrestricted, since it is not relying on the battery capacity.

In view of this, we propose that the use of UAVs (Unmanned Aerial Vehicles) be used to provide the additional channels required, in order to maintain an adequate QoS during these events. This has the benefit of the extra channels only being provided when needed; the system operation can return to the original number of channels afterwards. However, as in the case of selecting the adequate number of fixed channels, and using the ErlangB formula [13], we develop a mathematical analysis to carefully select these additional channels and obtain the ErlangU formula. Note that if a lower number of channels are used in the UAV, the blocking probability may not be lower than 0.02. On the other hand, if a lot more channels are placed on the UAV than those required to maintain a 0.02 blocking probability, the system would increase the operational costs due to the fact that more expensive UAVs would be required. In this work, we assume that the voice calls can be switched over to the aerial base station (UAV) and the terrestrial base station following a procedure similar to a handoff procedure in mobile cellular systems. To this end, there should be an active communication between the UAV and the base station to calculate power levels and demand voice channels whenever the drone is leaving the region of interest, with the following difference: In the case of a handoff in a cellular system, the mobile user usually does not know the exact moment of departure of a specific cell. Hence, in general, the handoff procedure is performed when the power received by a neighbor base station is much higher than the received power of the attending base station. However, in our case, the UAV can indicate, in advance, the terrestrial base station that will initiate the departure procedure some minutes beforehand (for instance, when the energy level in its battery is lower than a certain threshold), in such a way as to complete the handoff even before the UAV actually leaves the system. Also, the UAV may switch calls to the terrestrial base station when a channel in the latter becomes available. However, we do not further dwell on this issue; it will be further studied in future works. For example, different strategies will be proposed to initiate the handoff of calls being served by the UAV, including accepting/denying new calls to the UAV based on the energy level.

The organization of the paper is as follows: Section 2 discusses some previously published works on this topic; then Section 3 briefly describes the main characteristics of the UAV dwelling time, which is the time that it increases the system capacity. The design and mathematical model is detailed in Section 4. Section 5 shows some relevant results. We end this work with some conclusions and future works.

## 2. Related Works

In this section, we present the most relevant and previously published works on the use of UAVs in cellular networks and wireless communication systems to improve the system performance.

In [1], the authors study the use of UAVs for data sensing and then transmit them to a base station. This work explores the trade-off between keeping a certain *freshness* in the data and the communication tasks. However, the UAVs are not used to improve the performance of the cellular system; instead, the cellular system is used to transmit data from the UAVs.

In [12], the authors propose methods to recharge IoT nodes, and present a model for the energy consumption of the drones used to directly recharge nodes using different schemes. In [2] the authors also propose an energy consumption model focused on charging multiple UAVs according to a particular schedule. Also, in [3,14], the authors concentrate on charging drones; but, in this case, they propose using drones to charge other drones while in flight, in order to maintain a constant coverage of drones in a cellular system.

In [11], the authors study the use of UAVs to assist cellular systems with additional channels, in this case also using satellites for a backhaul connectivity. However, they do not consider the average dwelling times in the region of interest. Conversely, we focus on

the time that the UAV remains in the system, since these are the times that the blocking probability is reduced.

## 3. UAV Dwelling Times

There are many types of drones that can be used to assist the cellular system, such as the ones proposed in [12,15,16]. For drones in general, the service time, i.e., the dwelling time of the drone inside the region of interest, is determined by the energy drain of the battery. Recall that for our analysis, we only need to know the average time that the drone is inside the system, since this is the time that the system will have additional channels. The energy drain depends on the type of the drone, the environmental conditions, the flying pattern, and the activities of the drone inside the service area. In our case, we assume that a single UAV will travel from the charging station, which could be placed outside the serving area, to the region of interest, where the traffic load has an unusual peak activity; it will hover over this area, providing additional channels to users and then return to the charging station.

This assumes a flying pattern consisting of a vertical flight to ascend from the charging station, a horizontal forward flight, hovering in a steady state with no acceleration, and then a horizontal flight to return to the charging station, including a vertical flight to descend. The power required to perform these tasks is mainly related to the mass of the drone, vertical and horizontal speeds, and the thrust which is the required upward force. The energy consumption of the UAV can then be calculated according to the specific type of drone using the derived energy consumption models in the literature, such as the ones proposed in [2,12,15,16], among others. Also in [1], the authors propose an energy model for the energy consumption of the UAV that depends on the transmitting energy, the propulsion energy, and the hovering energy required in the operation of the system.

However, note that the energy consumption of the UAV is highly variable due to the environmental conditions and trajectories that the UAV must follow, which depend on the location of the traffic peak. Unlike [12], where the placement of nodes is known, in a cellular system, it is not predetermined the location where the traffic peak will occur, nor its duration. Also, these conditions may change during the UAV's flight at any point in time. As such, the use of energy consumption models, such as the ones discussed above, cannot be directly used to calculate or estimate the average dwelling time of the UAVs in the cellular system because the parameters, which are mainly constant values, have to now be random variables with an unknown distribution. In other words, in order to use the energy models reported in the literature, the times that the UAV consumed energy hovering and moving forward or backwards have to be statistically measured and known. Hence, we cannot directly use these equations, because the wind direction, trajectory, humidity, rain conditions, and many others are random variables that are unknown at this point.

Indeed, in order to calculate the average time that the UAV supports the cellular system with additional communication channels, it cannot be calculated directly from constant parameters, as reported before in the aforementioned energy consumption models. One alternative to the mathematical models would be the use of simulations to calculate the average dwelling time of the drone. However, this is not straightforward; we believe that this falls outside of the scope of this work, and we leave this research area for future works.

In [17], the authors present the analysis of two 3D models of cellular systems with UAVs: a truncated octahedron-based one and the binomial-Voronoi. The results show that the performance in terms of the coverage probability and average achievable rate are better in the first model, but the mathematical tractability and complexity are lower in the second. In [18], the authors analyze a network that uses UAVs to offload traffic from hotspots, considering that drones are connected to a power source from a ground station, and thus prevent the activation time of the drone from being conditioned by its energy consumption. The network performance is a function of the coverage probability, which depends on the length of the tether, the density of access points and the density of accessible buildings. In [19], the authors analyze a cellular system assisted by UAVs

based on stochastic geometry to find the coverage probability in both the access link and backhaul link. In particular, the downlink is analyzed considering that the distribution of the UAVs and the macro-base stations are modeled as two independent Poisson Point Processes, with line of sight links and non-line of sight links. The results show that the average performance depends strongly on the backhaul links, which are expected to be guaranteed. In [20], the authors simulate four frequency channel allocation algorithms in a multicellular network with the use of UAVs. The results show that the Forward-Looking Game (FLG) algorithm has a better performance in terms of the average capacity of the network when there are different values of SINR and the number of users. Note that these works have mainly focused on the coverage improvement provided by the drones but they have not studied the blocking probability reduction when drones are present in the region of interest.

In view of this, we develop a mathematical model that considers the average times that the UAV is assisting the cellular system, which directly depends on the energy consumption of the drones, and the times that the drone is outside this region, where the only available channels are the ones provided by the base station. Furthermore, we consider these dwelling times to be exponentially distributed. This assumption is twofold, firstly, because we do not know of any measurements regarding these times, and the exponential distribution may be used as a first approximation to this environment, and secondly, this model can be easily extended to consider different statistic characteristics of these dwelling times. Specifically, if these times are found to have a Coefficient of Variation (CoV) lower or higher than 1 (the exponential distribution has a CoV = 1), this approximation can be extended to the Erlang and Hyper-Exponential distributions, respectively. This would only require minor modifications to the exponential model presented in this work.

## 4. Teletraffic Analysis

In this section, we derive the analytical model to calculate the blocking probability of a UAV-assisted Cellular System. We now explain, in detail, the main assumptions and operation of the network. As shown in Figure 1, the main dynamics of the system are the following: The rate at which cellular users arrive to the system is $\lambda$. In this regard, users do not know (and have no way of knowing) if they are going to be served by a fixed terrestrial base station or by the mobile channels provided by the UAV. As such, note that the arrival rate is the same whether the UAV is inside or outside the region of interest. Similarly, the average call duration is considered to be the same, $1/\mu$, since, again, a user does not behave differently whether the call is served by a terrestrial base station or by a mobile base station. Then, users always leave the system at rate $\mu$. On the other hand, the UAV arrives to the region of interest with rate $\lambda_0$, which is related to two parameters: the time that it takes for the drone to reach the region of interest from the charging station and the average charging time. Indeed, we assume a single drone in the system that cannot arrive to aid the cellular system faster than the time it takes to completely charge its battery and travel to the zone where additional voice channels are needed. Also, the drone remains in the system for an exponentially distributed random time with parameter $\mu_0$, which is related to the average time that the drone can stay inflight, offering additional channels to the cellular system. Note that this time is related to the drone model (weight, type of motors, etc.), the environmental conditions, and the trajectory of the drone. From this description, we can observe that the presence/absence of the UAV device follows an ON/OFF process with rates $\lambda_0$ and $\mu_0$, that is, with an exponential duration of the ON (OFF) period with a mean value $1/\mu_0$ ($1/\lambda_0$).

Finally, the system model in Figure 1 also assumes that the cellular system is composed of s channels, while the drone has *M* additional mobile channels to increase the system capacity temporarily.

In this regard, we consider that it is composed of *S* servers (channels), which are fixed and installed at the base station. We also assume that the UAVs can support *M* mobile channels that are available to service users in the base station's coverage area; when

the UAV is flying inside the cell to provide enough channels to maintain the blocking probability in acceptable levels; and when the traffic load increases due to different events such as emergency, social, cultural, sportive, and political events, among others. In other words, we are considering scenarios where traffic arrivals increase in a temporary manner and return to normal levels in the following minutes/hours.

In view of this, we develop a Continuous Time Markov Chain (CTMC) to model the main dynamics of the system, namely, arrivals of users, departures of users, arrivals of UAVs, and UAV's departures, as depicted in Figure 1. Then, the CTMC is composed of two variables: $u$ and $i$. The latter corresponds to the number of users with an ongoing call in the system and the former corresponds to the case when the UAV is inside the cell's coverage area (adding $M$ channels to the cellular system). Then:

$$u = \begin{cases} 0; \text{ if there is no UAV} \\ 1; \text{ if the UAV is active in the cell} \end{cases} \tag{1}$$

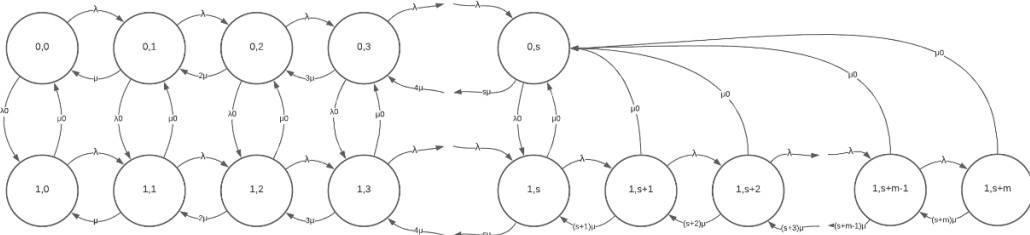

**Figure 1.** Markov Chain of the UAV-assisted cellular system.

From this description, the valid state space of the chain is described by: $\{\Omega_{u,i} : u = 1, 0; 0 \leq i \leq S + M\}$. As such, we can also write:

$$i = \begin{cases} \epsilon[0, S]; \text{ if there is no UAV } (u = 0) \\ \epsilon[0, S + M]; \text{ if the UAV is active in the cell } (u = 1) \end{cases} \tag{2}$$

Recall that the UAV can add $M$ additional channels; then, the number of available channels in the system is given by:

$$c = \begin{cases} S; \text{ if there is no UAV } (u=0) \\ S + M; \text{ if the UAV is active in the cell } (u=1) \end{cases} \tag{3}$$

which can also be written as $c = (1 - u)S + u(S + M)$.

The transitions of the chain are as follows:

- From state $(u, i)$ to state $(u, (i + 1))$: This occurs when there is a user arrival with rate $\lambda$ and $i < c$. Note that when the UAV is (not) serving the system, $i$ can take values up to $S + M$ ($S$).

- From state $(u, i)$ to state $(u, (i - 1))$: This occurs when there is a user departure with rate $i \times \mu$.

- From state $(0, i)$ to state $(1, i)$: This occurs when there is a UAV arrival with rate $\lambda_0 \times (1 - u)$. Note that when the UAV is serving the system, there are no further arrivals of UAVs. Indeed, we assume that a single UAV is serving the cellular system at a particular time. Otherwise, there should be active communication among UAVs or even swarms of UAVs which would greatly increase the complexity of the system; we believe this falls outside the scope of this work.

- From state $(1, i)$ to state $(0, i)$, when $i \leq S$: This occurs when there is a UAV departure with rate $\mu_0$. Note that, as in the case of UAV arrivals, there are no departures unless a UAV is inside the cell. In this case ($i \leq S$), the only channels used in the system are

provided by the serving base station. As such, when the UAV leaves, there are no dropped calls.

- From state $(1, i)$ to state $(0, i - S)$, when $i > S$: This occurs when there is a UAV departure with rate $\mu_0$, and there are users using channels of the UAV. Hence, when it leaves the system, all the calls being coursed by the UAV are dropped.

From this description, it is important to note that we assume UAVs remain in the system for a random exponential distributed time with mean $1/\mu_0$, and UAV arrivals occur in random inter-arrival times that are exponentially distributed with mean $1/\lambda_0$. Hence, we assume that the presence/absence of the UAV device follows an ON/OFF process, with an exponential distribution for both periods of activity and inactivity. As we mentioned before, the rationale behind this is twofold. First, this assumption allows us to simplify the analysis by considering a birth/death process for both the UAV and user's arrivals and departures. Second, to the best of our knowledge, there is no public data related to the dwelling and inter-arrival times of UAVs to increase the capacity of a cellular system. Hence, many measurements have to be conducted in practical scenarios in order to have an accurate statistical modeling of the UAV behavior, which most likely depend on the cellular coverage area, weather conditions, trajectory of the UAV, and obstacles in the serving cell, among others. In this regard, the exponential distribution can be considered as the base model, where the Coefficient of Variation (CoV) is equal to 1 (given the exponential distribution properties). For the case where practical measurements prove that the $CoV \neq 1$, an extension of this base model can be developed considering phase-type distributions (such as the Erlang, Hyper-Exponential or Coxian distributions) which are based on the exponential distribution and would only require minor modifications to the model presented in this work.

The aforementioned Markov chain is numerically solved to find the blocking probability as we describe it now. First note that, the first row of the chain corresponds to the case where the UAV is not active in the system. Hence, for all states in this row, there are only $S$ available channels. While on the second row, the UAV is active in the cell, and there are $S + M$ channels. The system goes from the first row to the second one when a UAV arrives in the system. Building on this, user arrivals only occur until state $(0, S)$ when the UAV is not in the system, and arriving calls are blocked, while in the case where the UAV is in the system, arrivals continue to arrive until the $S + M$ channels are occupied. Therefore, a user is blocked when in state $(S + M)$ if the UAV is serving the cell. Also note that if there are no more UAV arrivals once a UAV is already active, then they are only transitioning from state $(0, j)$ to state $(1, j)$ for $0 \leq j \leq S$; there are no more transitions from state $(0, S)$ to states $(1, j)$ for $j > S$ since the UAV is already in the system. Finally, note that when the UAV leaves the system, if there are channels of the UAV occupied (i.e., in states $(1, j)$ for $j > S$), the system goes to state $(0, S)$, dropping all those calls.

We now calculate the steady state probabilities, with $\pi_{u,i}$ using the rate equalization method [13], i.e., solving the following linear equation system. First, for the upper row, state (0,0), we have:

$$[\lambda + \lambda_0]\pi_{0,0} = \mu\pi_{0,1} + \mu_0\pi_{1,0} \tag{4}$$

For state (0,1), we have:

$$[\lambda + \lambda_0 + \mu]\pi_{0,1} = \lambda\pi_{0,0} + 2\mu\pi_{0,2} + \mu_0\pi_{1,1} \tag{5}$$

We follow a similar procedure until the last state of the first row, i.e., state (0,S), with the following equation:

$$[\lambda_0 + S\mu]\pi_{0,S} = \lambda\pi_{0,S-1} + \sum_{j=S}^{S+M} [\mu_0\pi_{1,j}] \tag{6}$$

For the second row, we have, for state (1,0):

$$[\mu_0 + \lambda]\pi_{1,0} = \lambda_0\pi_{0,0} + \mu\pi_{1,1} \tag{7}$$

For state (1,1), we have:

$$[\mu_0 + \lambda + \mu]\pi_{1,1} = \lambda\pi_{1,0} + \lambda_0\pi_{0,1} + 2\mu\pi_{1,2} \tag{8}$$

We follow a similar procedure until the last state of the second row, i.e., state (1,S+M), with the following equation:

$$[\mu_0 + (S + M)\mu]\pi_{1,S+M} = \lambda\pi_{1,S+M-1} \tag{9}$$

By numerically solving this linear system, we obtain the steady state probabilities $\pi_{u,i}$ for $u = 0, 1$ and $0 \le i \le S + M$. Hence, the blocking probability when the UAV is not serving the cell is $\pi_{0,S}$, while the blocking probability when the UAV is in the system is given by $\pi_{1,S+M}$. Since we consider exponential times for both the voice calls and UAV activities, we find the blocking probability based on the PASTA property. Specifically, the portion of the time that the system has all the channels occupied is the same proportion that the users have to view the system with no available channels whenever they arrive. This occurs at states $(0, S)$ and $(1, S + M)$ of the proposed Markov chain. Then, the total blocking probability is given by:

$$P_{B^U} = \pi_{0,S} + \pi_{1,S+M} \tag{10}$$

The probability that there is a UAV serving the system, $T_D$, is given by the average dwelling time of the UAV divided by the total time of the system operation, i.e., the average time that the UAV is not active (or in other words, the average time required for the UAV to arrive to the system), plus the average time that the UAV is active in the system. Then:

$$P[\text{There is a UAV}] = T_D = \frac{\frac{1}{\mu_0}}{\frac{1}{\mu_0} + \frac{1}{\lambda_0}} \tag{11}$$

and

$$P[\text{No UAV}] = 1 - P[\text{There is a UAV}] = \frac{\frac{1}{\lambda_0}}{\frac{1}{\mu_0} + \frac{1}{\lambda_0}} \tag{12}$$

On the other hand, the average number of calls supported by the UAV device, and consequently, the average number of calls that have to forcedly terminate their service when the UAV leaves the region of interest, can be calculated as:

$$N_D = \sum_{j=S}^{S+M} [(j - S)\pi_{1,j}] \tag{13}$$

Note that, in state $j$, for $j = S, S + 1, \ldots, S + M$, the UAV is always serving $(j - S)$ users that are forced to terminate their service when the UAV leaves the system. Also, in state $j(j < S)$ when the UAV leaves the system, we assume that all calls served by the drone can be accommodated to the terrestrial BS and no calls are dropped. From this, the drop call probability, i.e., the fraction between calls that are forced to finish and calls that are admitted, is given by:

$$P_{DA} = \frac{\mu_0 \times \sum_{j=S}^{S+M} [(j - S)\pi_{1,j}]}{\lambda \times (1 - P_{B^U})} \tag{14}$$

and the fraction of calls that are forced to finish and the total arrivals is given as:

$$P_{DO} = \frac{\mu_0 \times \sum_{j=S}^{S+M} [(j - S)\pi_{1,j}]}{\lambda} \tag{15}$$

## 5. Numerical Results

We now present the most relevant results derived by the analytical model presented above. The results presented in this section vary according to the number of mobile

channels (provided by the drone), $M$, and the drone's parameters, $a_D$, which captures the relation between the times that the drone is inside and outside the region of interest. From the system description, we can observe that our mathematical analysis does not require one to know the parameters of the drone, including thrust power, landing/takeoff, weight, type of motors, trajectory, environmental conditions, etc. Indeed, for a specific condition (with the combination of these parameters), we only require knowing the average dwelling time of the drone inside the cell, which is represented in the model in the variable $\mu_0$. Although the model assumes an exponentially distributed random dwelling time (which can be considered as an approximation to the real distribution), the developed model can be easily extended to consider other distributions. To the best of our knowledge, these dwelling times have not been reported before, and hence, we cannot evaluate the accuracy of this exponential assumption. In view of this, we only present analytical results. No simulations were performed, because we only evaluated the impact on the performance of the cellular system; considering specific drone or environmental conditions fall outside the scope of our work. Indeed, all of these parameters have an important impact on the average times that UAVs can remain in flight, which is the time that the cellular system will have additional channels to serve users in the region. Then, we vary $1/\mu_0$ in the range of [12,22] min (approximately [769,1346] s), considering also that the UAV may have to travel additional distance to reach the charging station and that many UAVs have an approximate flying time of 20 minutes. Hence, the value of $1/\mu$ only refers to the time that the UAV is inside the cellular system and does not consider the travel time from the charging station to the cell. We also consider that the average time between subsequent UAVs is $1/\lambda_0 = 16$ min (960 s), again considering that the UAV charging station may be placed outside the covering cell. Then, we introduce the parameter $a_D = \lambda_0/\mu_0$ that is analogous to the offered load of voice calls, i.e., the ratio of the arrival rate to the average call duration. In this case, $a_D$ does not represent real traffic to the system, but rather, it represents the ratio of the durations of the ON to the OFF periods. Following this, we call $a_D$ the *traffic load* of UAVs inside the cell, that is, the arrival rate of the UAV respective to its average dwelling time in the system. As such, the range of $a_D$ is [0.8, 1.4].

It is important to note that the proposed system can be used for cases where the traffic load increases temporarily but is expected to return to normal conditions. As such, we consider a traffic load, $a$, to be higher than the traffic load for which the system was designed. In other words, the system is designed such that the number of channels, $S$, in the base station can serve the users in the cell in peak traffic conditions with a blocking probability of 0.02 or less. Then, in our experiments, we consider traffic loads that entail higher blocking probabilities in the conventional system, i.e., in the ErlangB formula.

First, we focus on the blocking probability. To this end, in Figure 2, we show the ErlangB system, which does not consider any additional channels and is the conventional formula to calculate the blocking probability in a telephone system for voice services. We also show the ErlangU system, which was derived in the previous section for different traffic loads of users, $a = \lambda/\mu$, and the UAV traffic load, $a_D = \lambda_0/\mu_0$, in the ranges commented above. We can observe that the blocking probability for the conventional cellular system (i.e., with S fixed channels) remains constant for different values of $a_D$, since the dynamics of the UAV only provides additional channels for the UAV-assisted system. As such, the blocking probability only increases as the traffic load, $a$, increases. However, for the UAV-assisted system, as the UAV continues to have longer times in the region of interest (as the UAV continues to have longer times, $1/\mu$ increases, and then, $a_D$ increases), the blocking probability decreases considerably, since there is more time with additional channels to serve users.

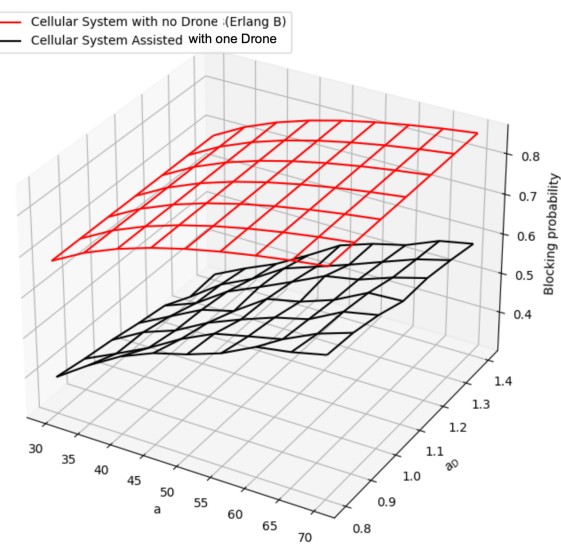

**Figure 2.** Blocking probability for the ErlangB system and ErlangU system for different users' traffic load, $a$ and UAV traffic load, $a_D$.

To further study the impact of the number of mobile channels supported by the UAV, we present, in Figure 3, the blocking probability for different numbers of channels added by the UAV, $M$. From these results, it is easy to observe the number of channels required to serve the cell by the UAV to increase the system capacity in cases of an increased traffic load, given certain events like earthquakes, sporting events, and others. Form these results, we can clearly observe that as the traffic load increases, we require more than 10 mobile channels provided by the UAV to maintain an acceptable blocking probability.

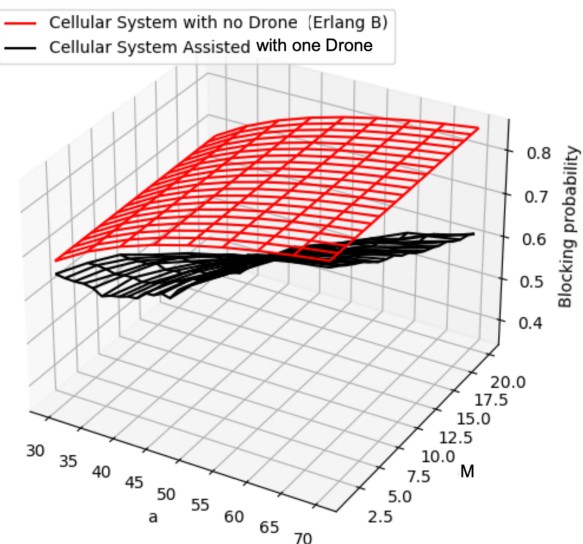

**Figure 3.** Blocking probability for the ErlangB system and ErlangU system for different users' traffic load and additional channels provided by the UAV, $M$.

In this regard, both Figures 2 and 3 offer a design tool for the deployment of such UAV-assisted cellular systems, in the sense that it is possible to determine the type of drone required in terms of the weight and battery capacity to provide specific flight times for the region of interest, as well as the number of mobile channels required to offer an adequate service to users in increased traffic load conditions. In other words, for a service provider that launches UAVs with mobile channels without considering these results may not significantly improve the system performance if the UAV does not leave

sufficient time in the region or if it does not carry the required mobile channels to reduce the blocking probability.

Now, we focus on the drop call probability. In this case, there are no drop calls in the conventional cellular system. In Figures 4 and 5, we show the fraction of the calls that are forced to terminate when the UAV leaves the cellular system, calls that are admitted, and the total number of calls that arrive, respectively, for different UAV *traffic loads*, as well as the number of channels provided by the UAVs. Clearly, the ratio of calls forced to finish to the total calls is higher than the ratio of calls forced to terminate to the calls that are admitted ($P_{DO} > P_{DA}$), but both can be used to determine an adequate Quality of Service to users under this proposed system, as well as to adequately select the type of UAV to be used in such applications. Considering that a dropped call has a very annoying effect on users with an ongoing call, these results allow a careful system design to keep this number as low as possible. Furthermore, this performance metric should be given preference over the blocking probability. Recall that a call is dropped during an ongoing call when the UAV leaves the system to return to the charging station, affecting the user's calls served by the UAV if the base station has no capacity to absorb such calls. Then, we can observe that as the number of mobile channels, $M$, increases, the number of dropped calls increases accordingly. The reason for this is that when all the channels at the base station are occupied and the drone is serving a high number of users inside the region of interest (high values of $M$), and when the drone must leave the area, all these users will have to terminate their call, increasing the drop call probability. However, at the same time, a high number of these mobile channels entails a low blocking probability. However, it is relatively insensitive to the value of $a_D$.

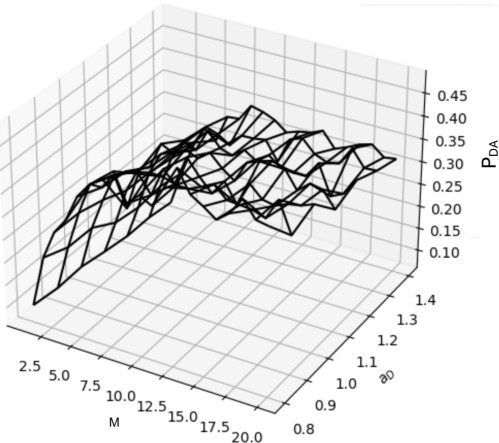

**Figure 4.** Fraction between calls that are forced to finish, and the calls that are admited for different UAV traffic load and additional channels provided by the UAV, $M$.

In Figure 6, we can see that as the traffic load of the drone ($a_D = \lambda_0/\mu_0$) increases, either by increasing the dwelling time or by reducing the inter-arrival times of the drone, the portion of time that the drone assists the cellular system, $T_D$ (given by Equation (11)), also increases. This result is relevant, since the dwelling time is mainly related to the battery capacity and energy consumption of the drone, while the inter-arrival time is given by the charging time and distance that the drone has to travel to arrive to the region of interest. The system administrator can assess whether a single drone would be sufficient for each specific case. In these results, we can also observe that the dwelling time of the drone is independent on the number of channels, $M$. This is because the dwelling time depends on the physical characteristics of the drone (weight, type of motors, batteries, etc.), the flight trajectory, and the environmental conditions.

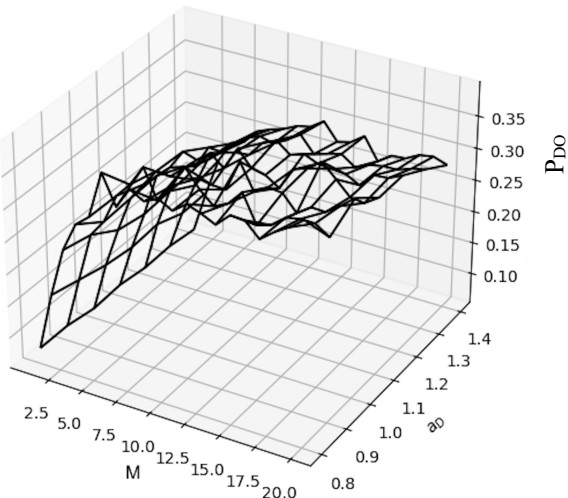

**Figure 5.** Fraction between calls that are forced to finish, and the total offered calls for different UAV traffic load and additional channels provided by the UAV, *M*.

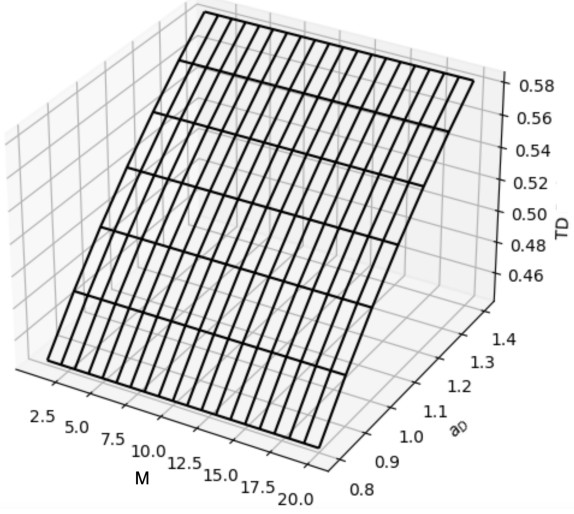

**Figure 6.** Portion of time that the drone is serving the cellular system for different UAV traffic load and additional channels provided by the UAV, *M*.

## 6. Conclusions

In this work, we analyzed, studied, and evaluated the blocking probability and average number of dropped calls of a UAV-assisted cellular system, considering that UAV temporarily provides additional channels whenever the traffic load increases in the coverage area. To this end, we develop the ErlangU formula that considers the main dynamics of the system, namely, arrivals and departures of voice calls, and the arrival and dwelling times of the drones. We provide clear guidelines for the design of the system by clearly calculating the number of fixed channels, *S*, provided by the base station and mobile channels, *M*, and also provided by the UAV to attain a target blocking probability. We can observe that as the number of mobile channels and/or the dwelling times of the drones increase, the blocking probability reduces. Hence, the system administrator can determine the adequate number of channels, capabilities of the drone to offer an adequate service, or even if more than one drone would be needed, reducing the average drone inter-arrival times. Also, we investigated the impact of the average dwelling times of the UAV and required time to reach the region of interest by introducing the UAV *traffic load*, $a_D$. This is a helpful parameter to adequately choose the UAV specifications, since the flight time is

directly related to the weight, types, and number of rotors, as well as the battery capacity. Indeed, if the UAV does not leave sufficient time in the region (with the derived results presented in this work, we can clearly observe the adequate times for different system conditions), the impact of additional channels may not be relevant.

As a future work, we intend to analyze UAV-assisted cellular systems using swarms of UAVs instead of the single UAV scenario, where the active communication and control between UAVs is required to avoid collisions and maintain the formation of the UAVs. Also, we plan to consider the case where UAV dwelling times in the system does not follow an exponential distribution. To this end, we will consider different effects such as wind and rain, as well as different types of trajectories of the UAVs inside the cell, that may impact the times that these devices can remain active in the system before they have to leave to recharge their batteries.

**Author Contributions:** M.E.R.-A. contributed with the conceptualization, methodology, validation, formal analysis and writing of the manuscript, I.V.-J. contributed with the conceptualization, formal analysis and writing (editing and review), I.Y.O.-F. contributed with writing, editing ,and formal analisys, N.T.-C. contributed with investigation and writing, A.P.R. contributed with writing, review and editing, and formal analysis. All authors have read and agreed to the published version of the manuscript.

**Funding:** This work was partially funded by project SIP 20240671 of the Instituto Politécnico Nacional.

**Institutional Review Board Statement:** Not Applicable.

**Informed Consent Statement:** Not applicable.

**Data Availability Statement:** This work is a theoretical work and no new data was created.

**Conflicts of Interest:** The authors declare no conflicts of interest.

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
