# Peer review of "Erlang-U: Blocking Probability of UAV-Assisted Cellular Systems"

_information, doi:10.3390/info15040192_

Round 1
Reviewer 1 Report (Previous Reviewer 3)
Comments and Suggestions for Authors
Some modifications are required.
1. The abstract should be written in a single paragraph.
2. In Fig. 4, when m increases, drop call probability also increases. explanation should be added.
3. In Fig. 5, Td should be explained. There is no meaning. Additionally, in the results, relationship between m and Td should be explained.
4. The parameters within the result figure should be expressed in terms of meaning rather than symbols such as m and a.
Author Response
We deeply thank the reviewer for the time and effort dedicated to the improvement of our work. In the attached file we detail the modifications and corrections derived from the reviewer's comments.

Reviewer 2 Report (Previous Reviewer 2)
Comments and Suggestions for Authors
See attached file.

Author Response
We deeply thank the reviewer for the time and effort dedicated to the improvement of our work. In the attached file we detail the modifications and corrections derived from the reviewer's comments.

Reviewer 3 Report (Previous Reviewer 1)
Comments and Suggestions for Authors
The paper will present a new approach to blocking a cellular system supported by UAVs (unmanned aerial vehicles) to temporarily increase the capacity of the communication network in the event of traffic congestion. The analysis presented in this work allows for the careful design of future communication systems requiring fewer channels that can serve users under normal traffic load conditions, while using UAVs to maintain an appropriate blocking probability when traffic load increases. To this end, we developed the ErlangU formulation, similar to the ErlangB formulation for a conventional cellular voice service system. The authors made corrections to the comments in the first review, re-analysed the results and improved the conclusions the article gained in substantive quality. The method discussed in the article is presented more clearly and contains the necessary technical data to replicate it, but I ask that the final decision to publish the article be made by the scientific editor.
Author Response
We deeply thank the reviewer for the time and effort dedicated to the improvement of our work. In the attached file we detail the modifications and corrections derived from the reviewer's comments.

Round 2
Reviewer 2 Report (Previous Reviewer 2)
Comments and Suggestions for Authors
After a third review, I have realized that the authors barely master the subject of teletraffic. For example, the PASTA property is misused in the context of the article. Likewise, in equation (14) \mu_0 should appear instead of \mu. So I consider that the work is far from offering minimum quality. I am not advocating accepting it, rather rejecting it.
Author Response
We have corrected the error pointed out by the reviewer and we thank you for your comments.

Round 3
Reviewer 2 Report (Previous Reviewer 2)
Comments and Suggestions for Authors
See attached file

Author Response
The reviwer mentions that it is implicitly assumed that when an ongoing call served by the base station ends, this channel is occupied by any ongoing call served by the UAV device, However, the proposed model does not directly contemplates this. The Markov chain should consider explicitly the number of channels served by the terrestrial BS and the number of channels served by the mobile BS and the proper transitions should be clearly described. However, our goal is not to study the switching procedure, but rather the blocking probability and drop call probability. As such, we do not agree that we assume a swtching of UAV calls to the terrstrail BS when a channel is freed in the terrestrial BS and the reviewer cannot make this assumption either, Alternatively, this model can also describe the system whe the assumption of not switching calls from the UAV to the terrestrial BS station when a channel is liberated in the former. However, these assumptions are not relevant for our system performance.
This is because expression (12) clearly states that in states \pi_(1,j), the number of channels supported by the UAV is (j-S) for j=S, S+1,...,S+M. As such, in these states, the UAV always serves j-S users ireespective of the swtching procedure. Hence, when the UAV leaves, there are j-S calls that are forced to finish their service. So, we maintain that this is the average number of calls that are dropped when the drone departures the system. Conversely, in states j<S, when the UAV leaves, we assume that all calls can be accomodated in the base statiion regardless of the moment when the switching procedure ocurred.
Finally, ErlangU comes from extending the ErlangB model to include the dynamics of a UAV, temporarily increasing the system capacity.
This manuscript is a resubmission of an earlier submission. The following is a list of the peer review reports and author responses from that submission.
Round 1
Reviewer 1 Report
Comments and Suggestions for Authors
This article will present a new approach to the problem of using mobile channels to help cellular systems increase network capacity for a limited period of time. In this paper, we determine the probability of blocking a UAV (unmanned aerial vehicle)-assisted cellular system to temporarily increase the capacity of the communication network in case of traffic congestion. The analysis presented in this work allows for the careful design of future communication systems requiring fewer channels that can serve users under normal traffic load conditions, while using UAVs to maintain an appropriate blocking probability when traffic load increases. The issues discussed by the authors of the article are currently being developed and refined. The article is interesting, but still requires the authors to address the technical parameters of the proposed solution (they need to give the technical parameters of the network and discuss in detail the system components used in Figure 1 with an explanation of the designations and add a description of the operation including the algorithms implemented in it and the conditions under which the tests were carried out. Please explain the parameters used in the formulae as some are missing. Please carry out more tests and describe them in detail, stating in which environment they were carried out and why such input parameters were chosen for the eclectic network. Please provide the efficiency of the proposed solution. In summary, there are several shortcomings in the paper such as:
1. The literature presented is incomplete and should be improved with more recent studies.
2. In the considerations related to the developed algorithm, please explain it better.
3. The article is missing the parameters of the equipment used in the tests whether these were computer simulations.
4. Please present more results and describe them in more depth regarding the effectiveness of the proposed solution.
5. Results and conclusions are sparsely discussed please improve this.
Reviewer 2 Report
Comments and Suggestions for Authors
See attached file.

Reviewer 3 Report
Comments and Suggestions for Authors
1. Some modifications are needed for the abstract. For instance, the abstract should be written in a single paragraph. Additionally, there is a need for revision in the first sentence as it lacks sufficient reason.
2. The second paragraph in the introduction needs to include units for the blocking probability.
3. Comprehensive revisions are required for the introduction. It should include a summary of related works. There is a need to address alternative approaches, such as the installation of temporary base stations using vehicles. The rationale for the necessity of additional base stations using UAVs should be explained. Additionally, an analysis of the advantages and disadvantages of using UAVs compared to existing approaches is essential.
4. In Chapter 3, the mathematical model proposed for UAV dwelling time cannot be found. It requires the addition of a mathematical model and an explanation.
5. An explanation of the background knowledge regarding the existing blocking probability is necessary.
